# A Model to Compare International Hospital Bed Numbers, including a Case Study on the Role of Indigenous People on Acute ‘Occupied’ Bed Demand in Australian States

**DOI:** 10.3390/ijerph191811239

**Published:** 2022-09-07

**Authors:** Rodney P. Jones

**Affiliations:** Healthcare Analysis and Forecasting, Wantage OX12 0NE, UK; hcaf_rod@yahoo.co.uk

**Keywords:** bed availability, international benchmarking, capacity planning, hospital bed numbers, new methods, integrated care, hospital bed modelling, hospital bed occupancy, indigenous health, Australia, Northern Territory

## Abstract

Comparing international or regional hospital bed numbers is not an easy matter, and a pragmatic method has been proposed that plots the number of beds per 1000 deaths versus the log of deaths per 1000 population. This method relies on the fact that 55% of a person’s lifetime hospital bed utilization occurs in the last year of life—irrespective of the age at death. This is called the nearness to death effect. The slope and intercept of the logarithmic relationship between the two are highly correlated. This study demonstrates how lines of equivalent bed provision can be constructed based on the value of the intercept. Sweden looks to be the most bed-efficient country due to long-term investment in integrated care. The potential limitations of the method are illustrated using data from English Clinical Commissioning Groups. The main limitation is that maternity, paediatric, and mental health care do not conform to the nearness to death effect, and hence, the method mainly applies to adult acute care, especially medical and critical care bed numbers. It is also suggested that sensible comparison can only be made by comparing levels of occupied beds rather than available beds. Occupied beds measure the expressed bed demand (although often constrained by access to care issues), while available beds measure supply. The issue of bed supply is made complex by the role of hospital size on the average occupancy margin. Smaller hospitals are forced to operate at a lower average occupancy; hence, countries with many smaller hospitals such as Germany and the USA appear to have very high numbers of available beds. The so-called 85% occupancy rule is an “urban myth” and has no fundamental basis whatsoever. The very high number of “hospital” beds in Japan is simply an artefact arising from “nursing home” beds being counted as a “hospital” bed in this country. Finally, the new method is applied to the expressed demand for occupied acute beds in Australian states. Using data specific to acute care, i.e., excluding mental health and maternity, a long-standing deficit of beds was identified in Tasmania, while an unusually high level of occupied beds in the Northern Territory (NT) was revealed. The high level of demand for beds in the NT appears due to an exceptionally large population of indigenous people in this state, who are recognized to have elevated health care needs relative to non-indigenous Australians. In this respect, indigenous Australians use 3.5 times more occupied bed days per 1000 deaths (1509 versus 429 beds per 1000 deaths) and 6 times more occupied bed days per 1000 population (90 versus 15 beds per 1000 population) than their non-indigenous counterparts. The figure of 1509 beds per 1000 deaths (or 4.13 occupied beds per 1000 deaths) for indigenous Australians is indicative of a high level of “acute” nursing care in the last months of life, probably because nursing home care is not readily available due to remoteness. A lack of acute beds in the NT then results in an extremely high average bed occupancy rate with contingent efficiency and delayed access implications.

## 1. Introduction

Germany, Austria, and the USA are usually recognized as having a high number of hospital beds [1]. However, is this a benchmark to which others should aspire, given the very small average size of a hospital in these countries [2]? How is it that Japan has relatively low numbers of critical care beds yet seemingly has one of the highest numbers of “hospital” beds in the world [1]? Is there some peculiarity in the way Japan counts “hospital” beds? [2].

How do we sort the fact from fiction and arrive at a true like-for-like comparison between countries? Up to the present, population age structure has been considered the primary factor in determining bed demand in which older age leads to higher demand [3,4] although various modifications are then added for various demand modification schemes and anticipated changes in future length of stay [5,6,7]. However, research indicates that the absolute number of deaths, irrespective of age at death, seems to play a far greater role than has been appreciated [3,5]. It is not widely appreciated that acute demand (and hence costs) increases with nearness to death (NTD) or time to death (TTD) far more so than with age per se [2,8,9,10,11,12,13]. Cognitive and neurological decline is also more related to nearness to death than to age [14], as are prescription costs [15] and the frequency of disability/disablement [16]. A composite score based on blood biochemistry showed only a slow decline with age but an experience of major and sudden deterioration in the last year of life [17]. Use of a hospital bed is especially related to nearness to death [18,19]. Most importantly, the demand for a hospital bed in the last year of life is approximately independent of the age at death [19,20]. In the developed countries, only around 10% of the population dies each year, and the 90% of persons not in the last year of life implies that mixed age and NTD models are required [15] to accurately quantify bed demand.

The NTD effect has been known since the 1960s, and economists have used the NTD effect over many years to investigate future health care costs [21]; however, this knowledge has not filtered through into the realm of health care capacity planning.

Based on the above, in 2010, this author called for hospital bed modelling to include the NTD effect [4]. This was followed by study in 2011, which noted that a simple ratio of occupied beds per death (all-cause mortality covering all places of death) remained remarkably constant over many years in both Australia and England [22]. This was despite the number of deaths being highly volatile from one year to the next [23]. In another study, the baseline trend for occupied medical beds per 1000 deaths remained unchanged over a 20-year period in England [24]. This seemingly contradicted the accepted wisdom that acute demand was largely age dependent.

Given this mixed dependence on age and NTD, a method that relies on both population age structure and nearness to death has been used to reveal the extent of bed number differences both between and within countries [1,2,25,26]. There is a logarithmic relationship between the ratio of beds per 1000 deaths and deaths per 1000 population.

This method seems to work equally well with available versus occupied beds and with total hospital beds (acute + maternity + mental health), acute-only, medical-only, and critical care beds [25,26,27,28], but the intercept and slope of this relationship are unique to the different measures of beds, i.e., occupied versus available, and for acute, medical, or critical care. Hence, the first aim of this study is to clarify the relationship between the slope and the intercept to improve the method and allow its extension to encompass a wider range of country- and area-specific factors.

The method allows rapid comparison, which then provides the basis for more detailed analysis behind the reasons for any differences. In most cases, the differences are due to the availability of funding either as GDP for countries, relative affluence between states, or issues of government resource allocation.

This relationship holds for three reasons:Firstly, arising from the NTD effect, some 55% of a person’s lifetime use of a hospital bed occurs in the last year of life—irrespective of the age at death [18].In addition, any agent that is capable of precipitating death, i.e., extremes of temperature, spikes in pollution, or outbreaks of infectious agents, will lead to multiple times more admissions than deaths [29]. COVID-19 has been a good example of this morbidity/mortality relationship.In more disadvantaged populations, people tend to die at a younger age; hence, age-based forecasting using national averages is unreliable. Death per se then becomes a more reliable measure of bed demand although more disadvantaged populations have slightly higher bed demand per death [18]. See Section 4.9 regarding social groups and bed demand.

One study suggested that around half of medical bed demand responded to the role of the absolute number of deaths, while the remaining half responded to the more traditional population age structure [29]. Pregnancy, childbirth, neonates, and paediatrics (especially in the first year of life) are examples of care that depends primarily on births or age structure. However, infectious agents prompting death in the elderly will also lead to elevated admissions in pregnancy and childhood [30]. Mental health conditions also seem to be exacerbated by infections [31,32].

Hence, while the new method was an empirical discovery, it is simple to use and seems to work in the real world. However, it is likely that there will be people groups within different countries that have higher than average demand for hospital beds.

From an international perspective, indigenous people around the world have unique health care needs [33], as do people from different racial/cultural backgrounds [34]. This study will use Australian indigenous people as an example of one such group and how this translates into acute bed demand.

It has been recognized for many decades that Australian indigenous people have health care needs that do not necessarily fit with a Western cultural and medical perspective [35,36]. Consultation with local indigenous people has led to improvements in health equity [37,38]. However, disparities remain especially among men [39], and access to surgical care is low [40]. Health literacy remains low [41], and levels of chronic and infectious disease remain high [41,42]. Higher levels of obesity compared to non-indigenous people remain a risk factor for poor health [43]. Consequently, 50% of indigenous people have risk factors for cardiovascular disease, and only 36% living with diabetes have blood glucose within guideline range [44]. Diabetes is a risk factor for coronary heart disease, hypertensive disease, and kidney failure [44].

Such health care needs should (in theory) translate into how health care resources are allocated in different locations. In Australia, capital spending is funded by state governments who are also responsible for capacity planning [45,46]. However, a proportion of state income is from the Australian Goods and Services Tax (GST), which is allocated to states based on a complex formula [47]. In the Northern Territory, planning is the remit of the Northern Territory Health and Hospital Services Council [48], which relies on funding from the NT Department of Health. Hence, all final capital allocations are left for each state to decide [45]. The Medicare funding formula is also known to significantly disadvantage the Northern Territory [49].

A recent study using this method identified that the Northern Territory (NT) and the Australian Capital Territory (ACT) had slightly more available total hospital beds than the Australian average, while Tasmania had fewer [25]. However, the Australian average implies average levels of remoteness; proportion of indigenous people and of hospital size, of which the latter directly impacts the average bed occupancy margin [50,51,52,53,54]; and hence the number of available beds.

In this respect, the Northern Territory (NT) has a 20-times higher proportion of population living in remote and very remote areas compared to the Australian average [55], a 9-times higher proportion of indigenous people [56], and a 4-times higher proportion of hospitals in remote areas [57].

In recent years, the provision of acute beds in the NT has received press attention with claims that hospitals are overcrowded and understaffed [58,59] and require more beds [60].

While available beds are a useful measure of hospital bed supply, it is the number of occupied beds that gives a measure of the expressed demand for hospital beds. This study will use the new method applied to occupied acute beds, i.e., excluding maternity, paediatrics and mental health, for Australian states to seek to determine if the NT has higher bed occupancy and therefore needs more or fewer beds than its current situation.

The revised method will also be used to re-evaluate a previous study regarding total occupied beds in English Clinical Commissioning Groups.

## 2. Materials and Methods

Data from previous studies have been re-analysed in this study [1,2,25,26,27,28]. Births and deaths in English and Welsh local authorities in 2019 were from the Office for National Statistics [61]. The number of patients waiting for an elective intervention in England was from NHS England [62]. Monthly deaths in England and Wales were from the Office for National Statistics (ONS) [63]. Occupied adult acute beds in Australian states in 2018/19 were from the Hospital Resources Table from the Australian Institute of Health and Welfare (AIHW) [64,65]. Same-day-stay admissions were assigned an 8 h stay (0.35 days) in the calculation of occupied beds, and this was added to the AIHW midnight occupancy data. Indigenous and non-indigenous deaths and population were from the Australian Bureau of Statistics (ABS) [66,67,68]. Indigenous and non-indigenous population by state was from the Australian Bureau of Statistics [66]. Occupied bed days and admissions during pregnancy and childbirth, ICD-10 Chapter O, in England were from NHS digital [69].

Occupied bed days for indigenous and non-indigenous groups associated with deaths and population were determined using the Solver simultaneous optimization function in Microsoft Excel. The Solver function was set to minimize the difference between the actual occupied beds and the predicted occupied beds. Note that occupied beds are occupied bed days divided by 365 (days per year). After a series of initial trials, the bed days associated with the non-indigenous population was set at 15 bed days per 1000 population (equivalent to 0.041 occupied beds per year per 1000 population) and the Solver optimization determined the final values. The spreadsheet with input data and equations is available on request.

## 3. Results

### 3.1. Defining Lines of International Equivalent Bed Availability

Earlier studies on this topic noted that the slope and intercept of the log (natural) relationship between beds per 1000 deaths and deaths per 1000 population were seemingly related.

A brief outline of the method is necessary. The basic logarithmic model was derived after investigating different ways to incorporate deaths and population in an international bed number data set, which was reported as beds per 1000 population. The logarithmic relationship arose as the best common mathematical function to fit the international data.

However, three different linear equations were proposed to describe the relationship between the slope and intercept for total hospital beds, medical beds, and critical care beds. Upon further investigation, it was realized that the three linear equations were part of a single non-linear relationship. In retrospect, the three linear relationships were linear approximations to the actual overall non-linear relationship.

This single relationship is illustrated in Figure 1, where it is assumed that the trend line goes through the origin. Each data point in Figure 1 is the result of an iterative process where the original data sets were re-analysed. Each value of slope and intercept is derived from visually grouping countries into roughly similar groups of “very high” down to the “lowest” bed availability. The value of slope and intercept for each group is recorded. Different countries that seemed to be outliers were then moved between groups and the revised slope and intercept recorded. The number of groups was limited by the available countries in each of the three studies. This was performed across multiple iterations. All data were then plotted on the equivalent to Figure 1, and outlying values from the iterative process were then removed, as were multiple examples of the values, which were almost identical. The final outcome of this process is the data shown in Figure 1.

Hence, the slope can be replaced by the intercept using the equation:Slope = 0.000008756 × Intercept^2^ − 0.29322 × Intercept(1)

This non-linear relationship means that an infinite series of lines of like-for-like bed provision or occupancy can be drawn based on the value of the intercept. Examples of these are given in the next section.

### 3.2. Occupied Beds in English CCGs Re-Evaluated

Figure 2 illustrates this relationship between slope and intercept (as above) using data relating to occupied total beds from English Clinical Commissioning Groups (CCGs).

In Figure 2, the CCG with the least occupied beds lies on the line of equivalence with an intercept = 620 beds per 1000 deaths, while a minority of CCGs lie above a line of equivalence with an intercept > 950 beds per 1000 deaths. It can also be seen that English CCGs have markedly different age structures ranging from a minimum around 4 deaths per 1000 population through to a maximum of around 14 deaths per 1000 population.

The major limitation of Figure 2 is that the occupied beds cover all bed types, namely adult and paediatric acute care, maternity and childbirth, and mental health. Paediatric, maternity, and childbirth all depend on the trends in births. Figure 3 therefore shows the ratio of births per death in English and Welsh local authorities. The year 2019 was used as the basis for Figure 3 to avoid the skewing of deaths in 2020 due to the COVID-19 outbreak.

One or more local authorities are contained in each CCG. From Figure 3, all the London boroughs (left-hand side of Figure 2) have very high ratio of births per death (>2.5 births per death), and this will imply that paediatric, maternity, and childbirth care will encompass a disproportionately high proportion of total occupied beds, and hence, these CCG tend to lie above the line of equivalence with an intercept > 950 beds per 1000 deaths.

### 3.3. Occupied Acute Beds in Australian States

An earlier study using this method contained a comparative study of available hospital beds in the states of Australia [1]. That study confirmed an already quantified available bed deficit in the state of Tasmania.

Appendix A shows a comparison of the indigenous versus non-indigenous people in the Northern Territory (NT) to the Australian average. From Appendix A, the NT has a 9-times higher proportion of indigenous people who have up to 6-times higher rates of admissions for certain conditions, have the highest indigenous death rate per 1000 population of all states, etc. Bed demand is therefore expected to be far higher in the NT than other states. Data for the 2018/19 financial year were used in this study to avoid the effects of the COVID-19 pandemic on hospital utilization during 2019/20.

The relationship from the previous Australian study lies within the usual scatter around the trend line, and hence, the conclusions from that study regarding available beds remain valid since Australia operates a hospital system very near to the international average for available beds. However, Figure 4 gives an appraisal of occupied adult acute beds in Australian states. The unmarked square (after Queensland) is the Australian average.

Application of this method to acute bed occupancy in Australian states reveals that the Northern Territory has a health care system that at first appears overly dependent on acute inpatient care (Figure 2). In Figure 2, we see from the *x*-axis that the Northern Territory and the Australian Capital Territory have the youngest populations and hence the fewest deaths per 1000 population. At the other extreme, South Australia and Tasmania have the oldest populations and hence the highest deaths per 1000 population.

On the *y*-axis, we have the ratio of occupied beds per 1000 deaths. As the trend line shows, younger populations have more occupied beds per 1000 deaths since there is less end-of-life care (fewer deaths) and more care devoted to younger people. The trend lines are defined by the intercept as per Equation (1). Tasmania lies below the trend line due to a long-recognized lack of acute beds [3] and, consequently, has 22% fewer occupied beds than the trend line. Queensland has the next largest deviation from the trend line of + 7% occupied acute beds and then Western Australia with minus 5%.

As can be seen in Figure 4, most states lie between the lines of constant bed occupancy with intercepts 750 to 800 beds per 1000 deaths. Tasmania not only has fewer “available” beds [1] but very low occupied beds. The Northern Territory seems to be a clear outlier with 86% more occupied beds than suggested by the intercept = 800 beds per 1000 deaths line of equivalence. By way of comparison, England with its ageing population has a ratio of 8.9 deaths per 1000 population. At the Northern Territory value of 4.5 deaths per 1000 population, England would have around 400 to 450 occupied acute beds per 1000 deaths (from Figure 2). This is well below the “normal” for Australian health care; however, indigenous Australians are virtually absent from the population of England.

One of the remarkable outcomes of this analysis is the similarity in the value of the intercept between England and Australia in Figure 2 and Figure 4. See Section 4.3 regarding the role of hospital size on the average bed occupancy rate.

### 3.4. Small-Area Deaths Show Unusual Trends

The Introduction gave the suggestion that local outbreaks of 3000 known species of human pathogens, carried by international air travel, may be leading to peculiar trends in small-area deaths. Evidence for this suggestion has already been presented for Australian states [70]. Figure 5 presents further evidence of peculiar trends using the far smaller local government areas in the UK. In a rolling/moving 12-month total, seasonality is removed since all four seasons are always included.

A rolling/moving 12-month difference compares two 12-month blocks that are immediately adjacent. Hence, the total deaths in April 2021 to March 2022 will be compared to the total deaths in April 2020 to March 2021. COVID-19 was far more active in the earlier period, hence the negative difference in Figure 5. The large peak for the upper quartile of 24% for the 12 month ending in February 2021 was due to the COVID-19 pandemic, which adversely affected many high-population-density locations during the first two waves. Figure 5 is therefore a simple way to show the volatility in deaths, while the upper and lower quartile show that some areas are affected more than others.

The lower quartile is bounded by local government areas with 1000 deaths per annum, which implies one standard deviation of Poisson randomness is equal to ±3.2%. The median size is 1500 deaths per annum, hence ±2.6%. Therefore, the year-to-year differences are dominated by systematic rather than random variation.

### 3.5. The Elective Waiting List in the English NHS Rises with Deaths

The magnitude of the English NHS waiting list is generally explained as an outcome of austerity measures imposed in 2010 following the 2008 financial crash [71]. However, few will be aware that prior to 2012, total deaths in England had been steadily declining from a maximum of 562,200 for the 12 months ending March 1976 and finally reaching a minimum at the 12 months ending January 2012, after which the total began to rise [24].

This was the outcome of the World War II baby boom, as the leading edge of the baby boom had reached the age of 66, which is the exact age at which the absolute number of deaths begin to rapidly rise, especially among males [72]. Actuarial estimates of the point in time at which deaths were predicted to rise again were not at all precise (due to multiple assumptions), and it is only in hindsight that the increase after 2012 became apparent. In the UK NHS, the size of the waiting list has always been determined by the difference between available funding and the expressed demand. Figure 6 therefore explores the possibility that rising deaths after 2012 unexpectedly contributed to cost pressures, thereby increasing the size of the waiting list. A rolling/moving average/total was used to avoid the effects of seasonality.

The time-series commences at the 12 months ending July 2008 when there were 470,500 deaths in that 12 months. Deaths then declined to 459,900 in the 12 months ending January 2012—where the waiting list also reaches its minimum value—and are now running at 518,500 for the 12 months ending March 2022 after peaking around 600,000 during the COVID-19 pandemic.

As can be seen in Figure 6 there is a reasonable correlation between the rolling/moving 12-month total deaths and the rolling/moving 12-month average number waiting. Following a government funded waiting list initiative, the number waiting reached a minimum of 2.3 million in January 2009 [62]. The waiting list then slowly increased to 2.36 million in January 2012, after which it began to rapidly expand [62]. As noted above, January 2012 marks the point of minimum deaths; hence, the start of the rapid increase corresponds to the point after which deaths begin to rapidly rise. Clearly the effect of deaths and the decision to admit to the waiting list will contain lags; however, the point has been demonstrated.

## 4. Discussion

### 4.1. History of the Model

Over many years there have been endless arguments regarding how many acute hospital beds are needed in different locations [73,74,75], within a context that reductions in acute bed numbers are a perceived route to healthcare cost containment [24,76].

While several methods/models are available to calculate future bed numbers [5,6,7,77], experience shows that the calculation is fraught with problems, hidden assumptions, and subtle pressure from the perceived policy ideal that still fewer beds will be needed in the future [3,76]. In addition, international health care systems have their own unique expressed demand for hospital beds where nursing home beds, same day care, care at home, hospital avoidance programs in the elderly, new technology, and integrated care can all substitute for acute beds [2,77,78,79,80,81].

It was realized that an independent method was needed, which could allow like-for-like comparison, which was free of policy-dependent views and opinions.

During the process of validating the new method, it has been consistently noted that the reliance upon deaths is far greater than could be expected from simple bed usage in the last year of life. This conundrum is reconciled by the fact that any agent capable of precipitating death is also capable of precipitating multiple times more hospital admissions. For example, every influenza death is associated with around 12-times more hospitalizations [82]. In mid-2021, in the USA, there were 580 COVID-19 hospitalizations per 100,000 population [83] compared to 187 COVID-related deaths per 100,000 population [84], hence 3.1-times more hospitalizations (with associated length of stay) than total COVID deaths. These ratios are higher for occupied bed days.

This will be repeated across all human pathogens, of which there are around 3000 known species [85]. Hence, the absolute number of deaths is acting as an effective proxy for the much wider morbidity/mortality pyramid. It is proposed that mini outbreaks of the 3000 known species of human pathogens are creating local surges in hospital admissions as a by-product of the inflammation and immune changes during infection, which can act to exacerbate existing and seemingly unrelated conditions [86]. High-volume air travel acts to rapidly transmit these pathogens to areas where they would not normally be found [87,88]—this has been recognized for many years although the far wider potential arising from 3000 species of human pathogen has not been fully appreciated.

The method was first developed arising from the authors understanding that deaths were serving as a proxy for hospital bed demand but that age structure of the population would also be important. International data on hospital beds per 1000 population and deaths per 1000 population (the crude mortality rate, which acts as a proxy for the age structure) are readily available and only require division of beds per 1000 population by deaths per 1000 population to give beds per 1000 deaths. A process of trial and error eventually resulted in the (natural) log relationship. The first study determined the slope and intercept for the international data set. The slope was assumed to remain constant, while the intercept was assumed variable. Next came the realization that the slope and intercept may be related and that countries could be grouped ranging from low bed numbers in the worlds’ poorest countries through to very high bed numbers in what where the world’s richest countries. These groups were somewhat arbitrary. This was then extended to data covering medical [27] and critical care beds [28]. It was then realized that a common relationship between slope and intercept may exist, hence Figure 1 in this study.

Each country can fall into the different groups for very different reasons. Japan is a very high country because it effectively counts nursing home beds as “hospital” beds [2] but has a definition of a critical care bed consistent with other countries [28]. Other countries have moved much psychiatric care away from a hospital into community care, and the average bed occupancy rate between countries can vary considerably for a variety of reasons [2].

Sweden looks to be the most bed-efficient country in the world and lies just below the intercept 640 line, once again, due to heavy investment in integrated care [10].

However, within the same country, there can be huge regional/state differences, of which the USA is an excellent example [89]. The USA does not offer universal health care to its citizens, and bed resources follow wealth rather than need. States such as Oregon, Vermont, and Idaho have very low provision down to below the intercept 450 line, which is in the moderate-low group, which includes less developed countries such as Burundi, Equatorial Guinea, and Swaziland [89]. Three “very high” states in the USA service wealthy populations and contain tertiary hospitals servicing nearby states as is also the case for Monaco.

### 4.2. Why Total Beds Is a Poor International Comparator

The revised method (Figure 2) revealed some interesting discrepancies in the study of total occupied beds in English Clinical Commissioning Groups (CCGs), namely that at above 8 deaths per 1000 population, the CCG data follow the revised lines of equivalence but seem to deviate below this level of deaths, i.e., in younger populations that mainly live in large cities.

Strictly speaking, the revised method does not precisely apply to pregnancy, childbirth, neonates, or to paediatrics—all of which depend on the trends in births. Births and deaths in UK local authorities are not strongly correlated (R-squared = 0.47) [61]. Based on data from the United Nations, the correlation between the crude death rate and crude birth rate among world countries is even worse (R-squared = 0.01).

For example, local authorities in the UK with 1000 deaths per annum can have anywhere between 450 to 4000 births per annum, while areas with 1000 births per annum can have anywhere between 250 to 2000 deaths per annum [61]. This helps to contribute to the scatter observed in Figure 2. Very clearly, births will be strongly related to both neonatal and paediatric admissions, which will further contribute to the scatter in Figure 2. Finally, in England the bed days associated with mental health “admissions” are reported at the date of “discharge”. Hence, a few long-stay mental health discharges for one CCG can unduly elevate the “apparent” beds per 1000 deaths. This helps to further contribute to the scatter observed in Figure 2

Unpublished research by this author shows that the method does not apply to mental health as well (geriatric mental health excepted); i.e., it best describes adult acute care. Less developed countries probably also contain very few if any dedicated mental health beds. Hence, below 8 deaths per 1000 population, for small areas, the trends in births and the role of age in non-death-related care probably begin to play an increasing role. Hence, the method seems to work best in large areas but requires further modification for small areas. This is consistent with the observation that two compartment models (one for age and local morbidity profiles, and the other for nearness-to-death) are required for areas of health care such as prescription expenditure [15]. However, use of the method for adult medical beds [27] and adult critical care [28] may be less affected by these considerations; however, see Section 4.8 for the role of social groups in the expression of small-area demand.

### 4.3. Relationship between Beds, Deaths, and Population

Rearranging the relationship between beds per 1000 deaths and deaths per 1000 population gives the following relationship for bed numbers, where a and b are constants, D = deaths (thousands), and P = population (thousands):Occupied beds = D × [a + b × ln(P) − b × ln(D)]

Since a > b and P > D, to a first approximation, occupied beds are directly proportional to deaths plus a small contribution from the natural log of total population. This confirms the earlier observation that the ratio of occupied beds per death remains relatively constant over many years [22,71].

Hence, for each location (country/state/region/small area), the observed occupied beds will be the sum of this relationship across each of the major ethnic groups or social groups (see Section 4.8) in that location. For Australia, this will be British, Irish, Italian, German, Chinese, Indigenous, and others (mainly Asian and European) [90]. Each ethnic group will have its own unique value of the intercept as in Figure 1.

### 4.4. Care in Rural and Remote Areas

The proportion of the population living in rural areas ranges from 0% in places such as Singapore and Hong Kong through to 86% in Papua New Guinea and Eritrea [91]. At 14%, Australia is close to New Zealand, Brazil, and Norway. The USA and Canada have 18% [91]. In the less populated Northwest of England, there is an inverse relationship between distance to a tertiary centre and treatment (surgery or chemotherapy) uptake [92]. Places such as the USA have 16% of the population more than 30 miles to the nearest hospital and 30 million people living beyond a one-hour drive from a trauma care hospital [93]. In contrast, even in the less populated Southwest of England, the majority live within a 30 minutes’ drive to the nearest general hospital [94].

Rurality has a major impact on hospital size, which in turn is a controlling factor in the average occupancy margin. Smaller hospitals are forced to operate at a lower average occupancy rate [50,51,52,53,54]; hence, places such as the Northern Territory need more available beds due to the generally smaller size of hospitals. The average occupancy margin is therefore specific to each individual hospital. The Australian average is 32% of available beds provided by hospitals with >500 beds. This ranges from 72% in the Australian Capital Territory to 0% in the Northern Territory and Tasmania [57,64]. See the next section for further discussion.

This exemplifies the statement that every health care system has its own unique demand for acute beds. Bed availability in Australia lies close to the international average [25]. New Zealand and Singapore have the lowest available bed numbers for developed countries, seemingly due to many years’ investment into integrated care backed by targeted government policy [1].

### 4.5. Effect of Hospital Size

It is widely reported in the literature that 85% average occupancy represents the optimum for whole hospital occupancy. The reality is that this is a complete “urban myth” for which there is no evidence [95]. The study of Bagust et al. [96] is often cited in support of this claim; however, this study only relates to a *single-specialty* bed pool containing 200 beds under the limiting assumption of negligible seasonality in admissions [96]. It cannot in any way be applied to whole hospital occupancy, which is entirely dependent on the size of the constituent bed pools withing each hospital [50,51,52,53,54]. A whole hospital figure of 85% occupancy only roughly applies to hospitals with more than 1000 beds [54]. This myth has unfortunately become so entrenched among hospital managers and policy makers that it is impossible to have a rational discussion on the topic [97].

Average occupancy far below 85% will widely apply to maternity, neonates, paediatrics, critical care, mental health, and any other small-specialized bed pool [53,98,99,100]. Occupancies higher than 85% can be achieved in elective day surgery units, where a waiting list can be used to schedule close to 100% occupancy during the hours that the day surgery unit is open [101].

One of the implications of size is that occupancy is related to turn-away, i.e., access block. The implications of turn-away are ambulances queuing outside the ED, delays to admission from the ED, higher hospital mortality rates, operational complexity, and staff stress [50,51,52,53,54].

The largest hospital in the NT is the Royal Darwin, which only has 363 beds [102]. This hospital serves as a joint district general and tertiary hospital. By way of comparison, in densely populated England, some 58% of hospitals have over 500 beds [2], which allows them to reap economy of scale [50,51,52,53,54]. The very high bed occupancy at the Royal Darwin [59,60] is a key indicator that bed supply is insufficient to the expressed demand for beds. High bed occupancy is associated with multiple types of inefficiency, patient harm, poor staff mental health, and access block [6,77,103,104].

A study on whole-hospital average occupancy suggested that at around 400 beds, a hospital such as the Royal Darwin should be operating at somewhere around 70% to 75% annual average occupancy [54]. This seemingly low average is required to optimize patient flow into the smaller bed pools that make up the entire hospital, i.e., paediatrics, trauma and orthopaedics, gynaecology, general and elderly medicine, critical care, etc. To attempt to operate above this threshold implies a high level of operational chaos.

### 4.6. Too Many or too Few Beds in the Northern Territory?

We need to ask the question: does the Northern Territory have too many or too few acute beds? Based on the comparison of available beds per 1000 population, the Northern Territory seems no different to any other State [25]; however, using occupied rather than available beds (Figure 4), the Northern Territory now appears as an outlier.

An undue emphasis on inpatient care can be excluded since the Northern Territory has 28% of its public beds devoted to same-day care compared to the Australian average of 13% [64,65]. One potential explanation lies in the observation that the Northern Territory has 49% of its beds in remote areas compared to the Australian average of just 3% [64,65]. The Northern Territory also has the highest proportion of Aboriginal people (approximately 9 times more than other states) with known higher health care needs (Table A1).

The Northern Territory is also known to have a relative deficiency of residential and nursing home care [105], which is disproportionately worse for Indigenous people. While aged care is considered to commence at age 65 for non-indigenous Australians, poorer Indigenous health means that Indigenous aged care commences at 50 rather than 65 [106]. The Australian average is that 80% of Indigenous people over the age of 50 live in non-remote areas; however, this will be reversed in the Northern Territory (Table A1). All the evidence points to a serious deficit of acute beds in the Northern Territory.

### 4.7. Unusual Trends in Deaths

Figure 5 gives evidence for unusual trends in deaths among UK local authorities. Actuarial forecasts for deaths are generally based on population age structure and always give smooth line trends. Deviation from smooth line trends are demonstrated in Australian states [70] and in much smaller UK local authorities in Figure 5. Notable points in Figure 5 are the 12 months ending in June 2015 compared to the previous 12 months ending in June 2014, when there was an upper quartile difference of 12% and a lower quartile difference of 6.5%. This has never been adequately explained other than an influenza outbreak, with significant antigenic shift in which influenza vaccine effectiveness went negative in many world countries [107,108]. Negative influenza vaccine effectiveness implies that the unvaccinated fared better than the vaccinated—an outcome that is only moderately common and not widely known.

The large peak in 2020 and 2021 was due to the COVID-19 pandemic, which was an identifiable (but new) pathogen. The different-shaped trends in Figure 5 cannot exclusively be explained by influenza outbreaks, and hence, it is proposed that the largely unexplained trends are due to one or more of the 3000 known human pathogens. Such outbreaks may be restricted to a single local authority or to smaller local geographies. Spread will depend on all the usual features of infectious transmission [109]. As highlighted in Section 4.1, all infectious outbreaks will lead to both higher admissions and higher deaths. Hence the number of deaths become a proxy for the higher admissions in the last year of life and the higher admissions from infectious sources [29]. Clearly, no one is attempting to screen for all 3000 pathogens, and hence, most mini outbreaks go unrecognized and unreported.

### 4.8. Location-Specific Volatility in Demand

The year-to-year volatility in demand depends on both size (via Poisson-based randomness) and local area volatility in environmental factors that will include infectious outbreaks. A large study using elective and emergency admissions, GP referrals, and ED attendances showed that the real-world volatility in all health care demand parameters is 2- to 3-times higher than that expected from size, i.e., Poisson-based variation [110]. All aspects of health care demand likewise show location-specific volatility in demand [23,111,112,113,114]. Figure 5 is another illustration of this reality that managing demand is simply more difficult in some locations than others. A recent four-part study suggested that the volatility in deaths is higher in high-population-density locations [29,89,115]. Significantly the volatility in healthcare staff sickness absence was also highest in high-population-density areas [115]. Disease transmission will be higher in high-population-density areas, which supports the notion that outbreaks of the 3000 known species of human pathogens makes a significant contribution to location-specific volatility.

The implication to hospital bed planning is that high volatility locations require a lower average bed occupancy to cope with the peaks in demand. Likewise, they experience larger troughs, which make managing staffing costs more difficult. This effect is almost universally ignored. In the English NHS, the unfortunate managers of hospitals in such higher volatility locations are simply accused of being “poor” managers; i.e., “if we could find better managers the problems would go away”.

### 4.9. Roles for Social Group versus Deprivation Score in Health Care Demand

The healthcare literature contains abundant examples of studies showing that social deprivation is linked to higher demand. These studies usually report deprivation in deciles or quintiles. The use of deciles/quintiles is necessary because at higher granularity, the supposed relationships become highly uncertain.

The principal reason is that health care demand depends far more on social group, with associated health care behaviours, than it does on deprivation per se. For example, a Bangladeshi community has different health behaviours than, for example, a Polish community living at equal levels of deprivation in the UK.

It turns out that social group is far more predictive of health care demand; however, it is rarely used. Social groups are constructed using the same tools used to construct consumer groups in marketing research [116]. In the UK, the smallest geography at which census data are collected is an output area. Each output area is determined based on its dominant social group [116], of which there are 76 groups.

Appendix B gives an overview of the difficulties of forecasting health care demand using pregnancy and childbirth as an example. Figure A4 shows that the trends in births are entirely dependent on social group. In addition, ED attendances and critical care admissions all depend on social group [117,118,119]. A further (unpublished) study showed that the proportion of persons attending the ED who go on to become an inpatient is entirely dependent on social group; i.e., after adjusting for social group, there is no detectable role for deprivation.

In Australia, indigenous people living in remote areas compared to cities would probably form two social groups.

### 4.10. When Policy Contradicts Reality

Having worked within the English NHS for over 30 years, I am aware of numerous examples of health and social care policy that contradicted reality. For example, in the 1990s, a policy called the Private Finance Initiative (PFI) was introduced to fund public capital projects. PFI was a prohibitively expensive way to fund capital expenditure, and rules were introduced to ensure that hospital projects represented value for money [120,121]. Alas, the only way to achieve the value for money criteria was to “fiddle” future bed demand such that new hospitals were all smaller than those they replaced [122,123]. Large management consultancies were employed to produce so-called “robust” estimates for bed numbers that met the value for money criteria. All manner of dubious assumptions regarding future length of stay and admissions were made to justify these numbers [24,124,125]. Somewhat unsurprisingly, by 2019, the average acute hospital occupancy had risen to 92% in Q3 of 2019/20 (just before COVID-19) from 85.3% in Q2 of 2011/12 [126]. Such high levels of occupancy then create access blocks leading to long waiting times in the emergency department, as patients either wait in an ambulance to enter the ED or wait in the ED to access an inpatient bed [127,128].

This well-intended but misplaced policy and policy guidance has effectively crippled the English NHS.

The second example is a change in NHS funding precipitated by the 2008 financial crash. A decade of NHS “austerity” funding was implemented in 2010 [129]. It is often claimed that the flatline funding, which rose in line with inflation, was responsible for this increase in the waiting list [129]. Since 2012 (two years after the introduction of austerity), the number of persons waiting for an elective operation in the English NHS has risen dramatically, as shown in Figure 5. However, The Department of Health and Social Care in England views age as the sole factor responsible for costs. This is not an explicit policy but rather a firmly held view, which contradicts decades of research by economists [21].

The key observation is that the waiting list rapidly expands after 2012, i.e., after deaths have reached a minimum. Hence, before 2012, NHS cost pressures are partly mitigated by decades of falling total deaths. Suddenly, this source of cost pressure mitigation was removed, and the resulting financial imbalance was reflected in the rising waiting list [130].

### 4.11. Is Better Bed Modelling Possible?

Appendix B presents a case study relating to births and length of stay during pregnancy and childbirth, which illustrates the practical difficulties of forecasting future admissions and associated length of stay. This can be replicated across all acute specialties where the future is highly uncertain. Heroic assumptions about future admissions and length of stay can be almost guaranteed to vastly underestimate future bed demand.

A recent study has suggested ways in which better bed modelling can be accomplished [131]. No single model will give the “correct” answer and a range of models should be employed to triangulate the resulting outputs. The method discussed in this study can be used to sense check the outputs from these models [132].

## 5. Key Recommendations

Several key recommendations arise from this work, namely:Further work on international bed number comparisons should include both available and occupied bed numbers. Available bed numbers in isolation can be highly misleading.Bed numbers covering pregnancy and childbirth, neonates, and paediatrics should be reported separately to acute adult bed numbers.Bed numbers in the Northern Territory and Tasmania require urgent investigation. It is suspected that some of the issues behind Federal capital allocations and Medicare funding may be the root cause but are beyond the control of the state governments.Occupied beds in English CCGs should be re-investigated after excluding pregnancy and childbirth, neonates, and paediatrics and mental health.Mixed models covering age and deaths should be developed to better forecast bed demand. The outputs from alternative models should always be studied.Models using social groups should be further developed.The international model discussed in this study should be used as an independent way of validating the output from other models.While “political’” pressure to model the lowest possible bed numbers is undesirable, avoiding such pressure may be a theoretical ideal.

## 6. Limitations of the Study

All methods to estimate bed numbers contain limitations and hidden assumptions. All models tend to fail at the extremes—where data are scarce and where correlation methods tend to gravitate to whatever factors regulate demand nearest to the “average”, hence the obligatory need to compare the output from a range of models using different assumptions.

This is an empirical model that assumes a logarithmic relationship between the parameters. The logarithmic relationship was chosen because it gave the best fit from a range of common mathematical functions. It is possible that another mathematical relationship gives a (slightly) better fit to the data—although, as pointed out, maternity, paediatrics, and mental health should be excluded and modelled separately.

The study relating to the Northern Territory has the limitation that only a single year of data was analysed for the last year in the pre-COVID era. Data on indigenous/non-indigenous deaths for the Australian financial year were not available and were therefore estimated from a calendar year total. Since deaths appear in both axes, this implies that a confidence interval bubble will surround the data for each state. This confidence interval bubble will be highest for the two smallest states, namely Tasmania and the Northern Territory. This implies that both states will need a higher occupancy margin to cope with uncertainty in demand arising from environmental volatility—meteorological, pollutants, and infectious agents [133], hence the well-recognized need for more medical beds during the winter months.

The final limitation is that a very simple model (Indigenous versus non-Indigenous) was applied to the Australian data, and further modification will be required by Australian researchers. As proposed, modification by social group is highly recommended. In this respect, most countries have their own equivalent to the UK’s output area classification (OAC).

## 7. Conclusions

In conclusion, the pragmatic method has been further refined to give an infinite set of lines of equivalence that can be set by the intercept. Countries lie along lines of equivalence for a wide variety of reasons; however, countries with an efficient “universal” health care system seem to lie around the intercept 640 line. It is recommended that further studies on this topic separate out data on adult acute care and focus on occupied beds rather than available beds. Occupied beds can then be turned into available beds based on hospital size or country specific average occupancy—which depends on hospital size.

The new method provides a pragmatic and rapid way to assess relative bed demand between larger areas such as states. It works in the real world simply because death is serving as a proxy for both the nearness to death effect and the wider mortality/morbidity pyramid. The method reflects the known shortage of beds in Tasmania and additionally reveals that the Northern Territory has a unique health care system reflecting higher demands upon acute beds due to a large and dispersed (remote) indigenous population. This is further amplified into available beds due to the low average size of hospitals in the Northern Territory and a consequent enforced lower average occupancy margin.

## 8. Footnote

For those wishing to explore this area in greater detail, a list of over 200 publications is available covering forecasting demand, bed planning, roles for deaths in demand, and costs (including financial risk): (PDF) A collection of over 200 papers relating to health service research in the area of forecasting demand, financial risk, and the calculation of optimum hospital bed numbers and occupancy (researchgate.net, accessed on 1 August 2022).

## Figures and Tables

**Figure 1 ijerph-19-11239-f001:**
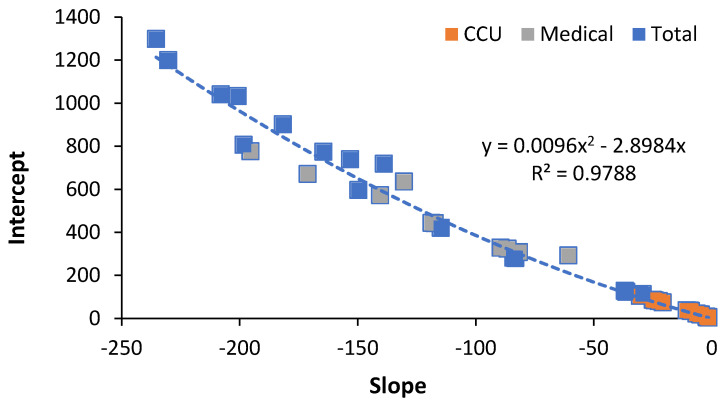
Relationship between slope and intercept for hospital bed numbers (critical care, medical or total beds) in world countries. Data are from three previous studies [25,27,28].

**Figure 2 ijerph-19-11239-f002:**
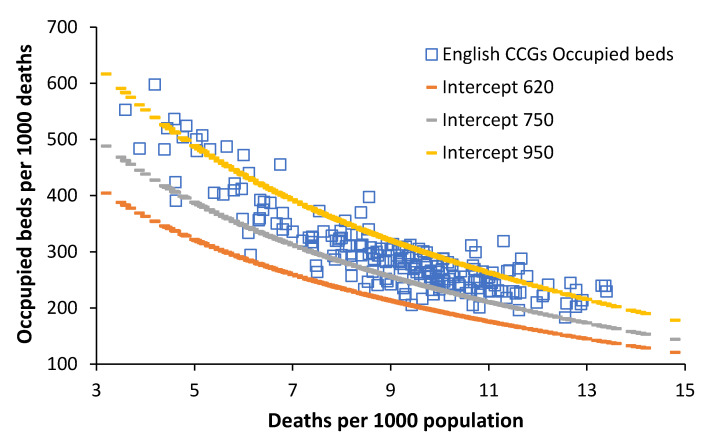
Lines of equivalent bed provision based on the intercept for English Clinical Commissioning Groups (CCGs).

**Figure 3 ijerph-19-11239-f003:**
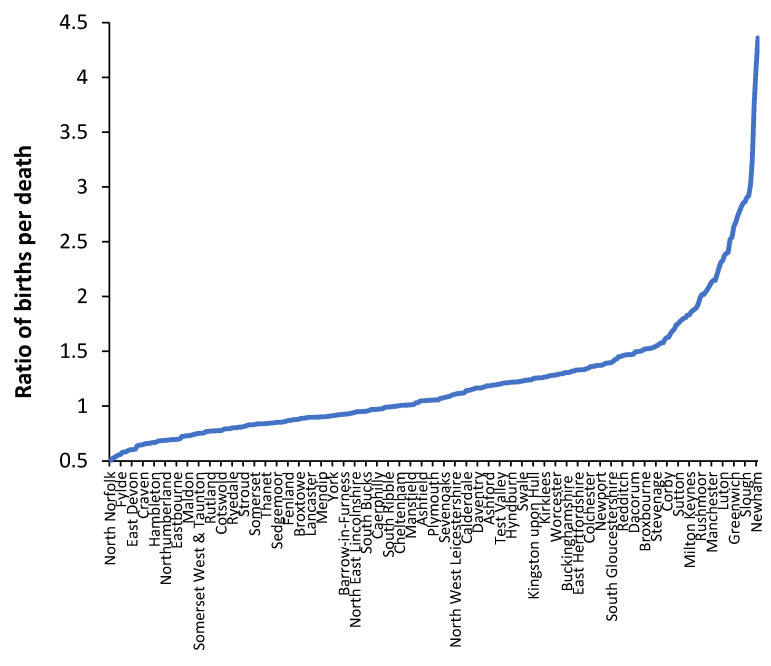
Ratio of births per death for English and Welsh local authorities in 2019.

**Figure 4 ijerph-19-11239-f004:**
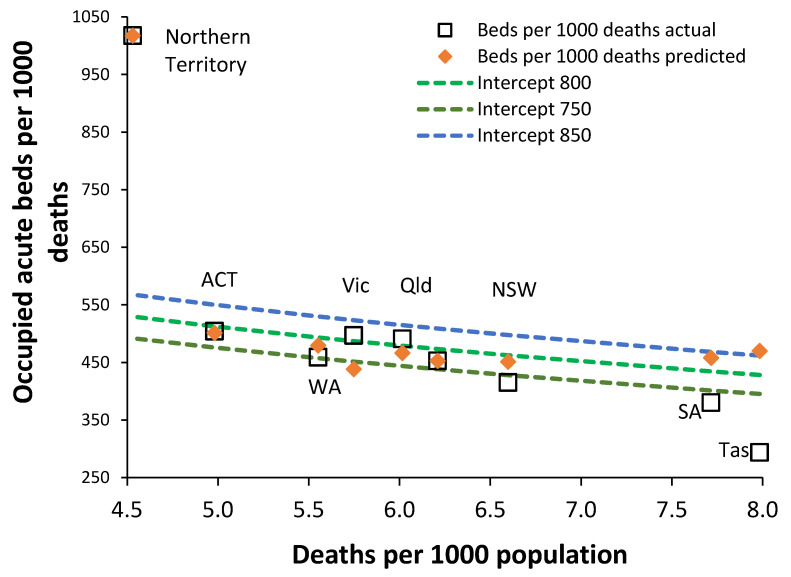
Occupied acute beds in Australian states compared using a new method for bed comparison. Data for occupied acute beds, i.e., excluding maternity and psychiatric/mental health, in 2018/19 are from Australian Institute for Health and Welfare [5] while data on State deaths and population in 2018 are from the Australian Bureau of Statistics [6]. ACT, Australian Capital Territory; WA, Western Australia; Vic, Victoria; Qld, Queensland; NSW, New South Wales; SA, South Australia; Tas, Tasmania.

**Figure 5 ijerph-19-11239-f005:**
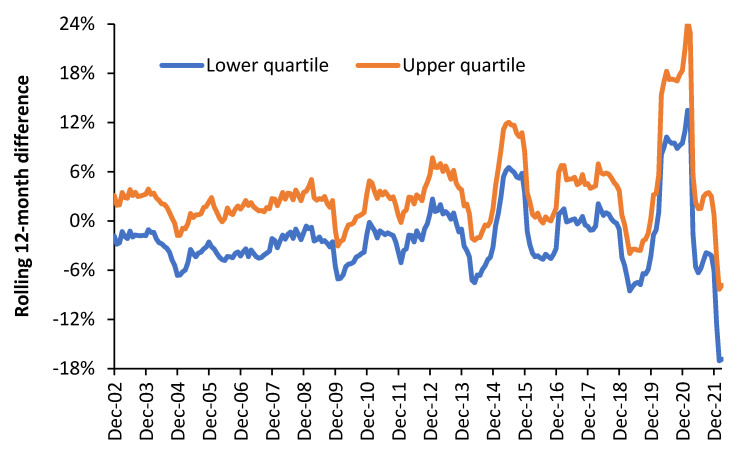
Lower and upper quartile for the rolling/moving 12-month difference in total deaths (all-cause mortality) in 516 UK local government areas and regions, 2001 to 2022.

**Figure 6 ijerph-19-11239-f006:**
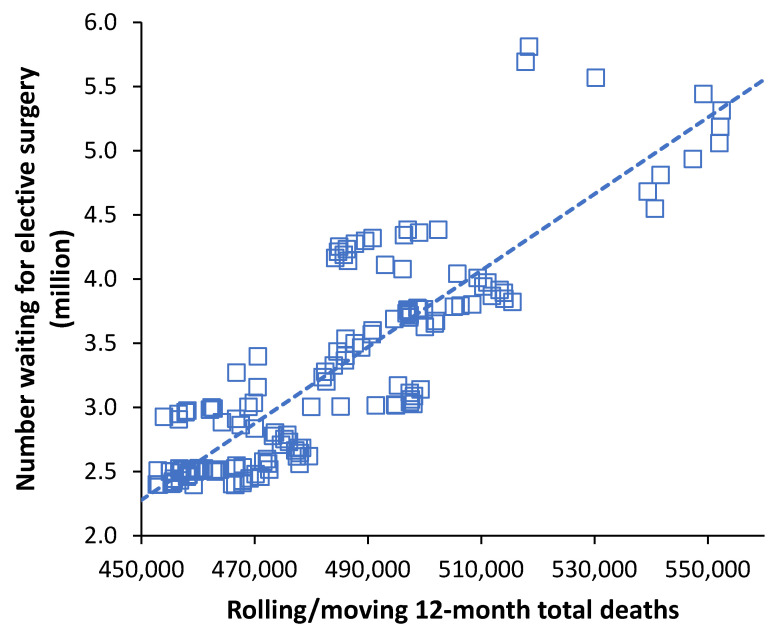
Rolling 12-month total persons waiting for elective surgery in England versus a rolling 12-month total of deaths. Data on the number of persons on the elective waiting list are from NHS England [62].

## Data Availability

All data are from publicly available sources.

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
