# Peer review of "A Model to Compare International Hospital Bed Numbers, including a Case Study on the Role of Indigenous People on Acute ‘Occupied’ Bed Demand in Australian States"

_ijerph, 2022, doi:10.3390/ijerph191811239_

Round 1

Reviewer 1 Report

The author presents a model comparing beds across the UK and Australia. The study findings are interesting and can impact health services planning. Kindly see the comments below for your consideration: 

1. Although the author reports on the methods employed, it is unclear how the analysis was handled. The author needs to provide a section on this. 

2. Ethics approval not mentioned. Though the author notes that this is secondary data analysis, it is important to specify if ethics approval was required for the initial studies. If this is not required, there should be a statement to highlight this. 

3. The key recommendations are particularly helpful as it translates all the model to clinical practice. Well done for this. 

4. There is no mention of limitations of the study. I think this is equally important to mention. 

5. As a minor comment, the English language can be improved further. 

Author Response

Many thanks for your time and comments.

The response to your questions are as follows.

  1. No ethics approval was required at any stage of data collection or analysis.
  2. Given the fact that there is no prior method to rely on the lines of constant bed availability/need were determined by trial and error. Data for different countries were grouped by eye and the line of best fit applied. Move data which seems to be marginal and repeat. I kept track of all the outcomes to arrive at the resulting data, after excluding itterative outcomes which were substantially away from the trend line. There may be some elegant mathematical method to do this but the final trend line seems to do the job.
  3. A section on limitations has been added which highlights the often unacknowledged fact that all methods contain hidden assumptions. Indeed, all models tend to fail at the extreemes. Hence my recommendation that multiple methods be compared.

Once again many thanks for your time and helpful comments.

Reviewer 2 Report

The author has a solid theoretical framing and debunks the urban myth of available beds, also emphasizing the atypical case of Japan. The fact that it has the highest number of beds is actually an artifact.

The highest strength of the current research is that the author proposes a new statistical method that is much more appropriate for cross-country studies.

More precisely, the method consists in plotting the number of beds per 1,000 deaths versus the log of deaths per 1,000 population. The statistics used are clearly explained and make a consistent contribution regarding the estimates on the basis of which specific public health policies will be developed.

The proposed method was even applied in a case study conducted on hospital situation in Australia. Disparities were highlighted between Northern Australia and Tasmania or the southern area, where the population is older and indigenous.

Good luck to the author in future studies that will now have an adequate technique to make scientifically valid comparisons.

Author Response

Thank you for your time and kind comments. I have been investigating hospital bed modelling for 30 years and this is the best 'bullet proof' model I have been able to come up with. The model was actually discovered by trial and error when I was attempting to analyse an international data set.

I have added the comment that mortality also incorporates the effects of deprivation/poverty (as will be reflected in social group) which is known to be associated with higher acute admissions.

Once again, many thanks.